# Direct Regulation of DNA Repair by E2F and RB in Mammals and Plants: Core Function or Convergent Evolution?

**DOI:** 10.3390/cancers13050934

**Published:** 2021-02-24

**Authors:** Swarnalatha Manickavinayaham, Briana K. Dennehey, David G. Johnson

**Affiliations:** 1Department of Epigenetics and Molecular Carcinogenesis, The University of Texas MD Anderson Cancer Center, Smithville, TX 78957, USA; SManickavinayaha@mdanderson.org (S.M.); BDennehey@mdanderson.org (B.K.D.); 2Center for Cancer Epigenetics, The University of Texas MD Anderson Cancer Center, Houston, TX 77030, USA

**Keywords:** E2F1, ATM, TopBP1, BRG1, p300/CBP, homologous recombination

## Abstract

**Simple Summary:**

Retinoblastoma (RB) proteins and E2F transcription factors partner together to regulate the cell cycle in many eukaryotic organisms. In organisms that lack one or both of these proteins, other proteins have taken on the essential function of cell cycle regulation. RB and E2F also have important functions outside of the cell cycle, including DNA repair. This review summarizes the non-canonical functions of RB and E2F in maintaining genome integrity and raises the question of whether such functions have always been present or have evolved more recently.

**Abstract:**

Members of the E2F transcription factor family regulate the expression of genes important for DNA replication and mitotic cell division in most eukaryotes. Homologs of the retinoblastoma (RB) tumor suppressor inhibit the activity of E2F factors, thus controlling cell cycle progression. Organisms such as budding and fission yeast have lost genes encoding E2F and RB, but have gained genes encoding other proteins that take on E2F and RB cell cycle-related functions. In addition to regulating cell proliferation, E2F and RB homologs have non-canonical functions outside the mitotic cell cycle in a variety of eukaryotes. For example, in both mammals and plants, E2F and RB homologs localize to DNA double-strand breaks (DSBs) and directly promote repair by homologous recombination (HR). Here, we discuss the parallels between mammalian E2F1 and RB and their *Arabidopsis* homologs, E2FA and RB-related (RBR), with respect to their recruitment to sites of DNA damage and how they help recruit repair factors important for DNA end resection. We also explore the question of whether this role in DNA repair is a conserved ancient function of the E2F and RB homologs in the last eukaryotic common ancestor or whether this function evolved independently in mammals and plants.

## 1. Introduction

The RB tumor suppressor gene (*RB1*) was originally isolated through positional cloning of a chromosomal segment frequently deleted in the childhood cancer retinoblastoma [1]. *RB1* is also commonly mutated in several other cancers, including sarcomas, lung cancers, and triple-negative breast cancers [2]. In many cancers, the RB protein is maintained in an hyperphosphorylated, inactive state by genetic or epigenetic alterations resulting in increased and inappropriate cyclin-dependent kinase (CDK) activity. Overall, RB function is disrupted in most, if not all tumors, and its disruption is considered a hallmark of cancer [3].

An important advance in understanding the molecular function of RB was the discovery that it is a component of complexes containing E2F family members [4]. E2F, a cellular transcription factor, was first characterized as a regulator of the adenovirus *E2* gene [5]. Subsequently, E2F1 was isolated as an RB-binding protein capable of activating the expression of both the adenovirus *E2* gene and endogenous cellular genes involved in DNA replication and cell cycle progression [5,6,7]. Additional RB and E2F family members were later identified by their sequence similarity in humans, other animals, and plants [8,9,10,11,12,13,14,15]. RB family members are identified by a conserved pocket domain, which binds to E2F proteins. E2F family members are identified by their DNA binding domains, which recognize similar DNA consensus binding sites in the promoters of cell cycle-regulated genes [13,15,16,17]. These E2F target genes encode proteins important for DNA replication and mitosis and are actively repressed when RB is bound to E2F.

In addition to regulating cell proliferation, both animal and plant E2F and RB homologs have important functions outside the mitotic cell cycle, including transcription-independent functions that maintain genome integrity and promote DNA repair. The role of these non-canonical functions of E2F and RB in suppressing tumor development and how the loss of these functions might be therapeutically exploited have been discussed in several recent review articles [18,19,20,21]. This review focuses on another open question in the RB/E2F field: Which shared cell cycle-independent functions form the original core functions of E2F and RB and which functions independently evolved later in multicellular animals and plants?

## 2. Evolutionary History of E2F and RB Homologs in Eukaryotes

Genomic sequence comparisons have revealed homologs of *E2F* and *RB* in all major groups of eukaryotes, indicating that these two genes were components of the original cell cycle network that existed in the last eukaryotic common ancestor (LECA) approximately 1.6 to 2 billion years ago [22,23,24,25]. The LECA likely had at least three E2F genes: One typical *E2F* gene, one atypical *E2F* gene, and one *DP* (dimerization partner) gene, each with related DNA binding domains [22,24,25]. Typical E2F proteins, such as human E2F1-E2F5, are dynamically regulated through interactions with RB pocket proteins and require dimerization with a DP subunit to efficiently bind DNA (Figure 1A). Atypical E2F family members, such as human E2F7 and E2F8, have two DNA binding domains, do not interact with RB homologs, and do not dimerize with DP proteins. Typical E2F family members generally repress transcription when associated with RB and activate transcription when free from RB. Atypical E2Fs repress the transcription of target genes by mechanisms independent of RB [9,26].

The LECA cell cycle network also likely included at least one CDK and multiple cyclins that regulated events in different phases of the cell cycle (Figure 1B). Studies from a variety of eukaryotes have demonstrated how RB and E2F homologs link the activity of CDKs with the periodic expression of genes important for cell cycle progression and cell division [13,17,27]. Phosphorylation of RB by D-type cyclin-CDKs initially allows the induction of genes important for the G1/S phase transition, including G1/S phase cyclins, like human *CCNE* (cyclin E) and *CCNA* (cyclin A). The accumulation of these cyclins further increases CDK activity in the cell, maintains RB in a hyperphosphorylated state, and allows for the E2F-dependent activation of genes important for DNA replication and mitosis (Figure 1).

Based on the genomic sequences of *S. pombe* and *S. cerevisiae*, it was originally thought that all fungi lacked homologs of E2F and RB [22,23]. In these yeasts, functions analogous to those of E2F-RB in regulating the periodic expression of genes important for cell proliferation have been taken on by the SBF/MBF family of transcription factors [23]. Further, just as E2F binding to RB prevents it from activating its target genes, SBF binding to partners such as Whi5 also blocks SBF from activating transcription, and just as RB is targeted by CDKs, Whi5 is also targeted for inactivation by CDKs [28,29]. E2F and SBF/MBF transcription factors also recognize similar DNA sequence motifs in the promoters of cell cycle-regulated genes [24]. Yet, despite recognizing similar motifs, the DNA binding domains of E2F and SBF/MBF transcription factors are unrelated and bind to DNA through different mechanisms [30,31,32]. The DNA binding domains of fungal SBF/MBF factors are most closely related to the KilA-N domain-containing proteins found in a family of DNA viruses that infect both eukaryotes and bacteria [24,33].

Although absent in *S. pombe* and *S. cerevisiae*, some fungi have E2F and RB homologs. Zoosporic fungi, which include the flagellated chytrids, have a hybrid cell cycle network in which both E2F-RB and SBF-Whi5 factors regulate the expression of genes involved in cell proliferation [24]. In other types of fungi, such as common bread molds, RB homologs are absent but E2F homologs are present. This suggests a model in which the ancestral fungal progenitor bore homologs of E2F and RB that were lost in specific linages as SBF-Whi5 replaced E2F-RB in the fungal cell cycle network (Figure 2A). This model is supported by phylogenic analyses demonstrating that plants, which have a full complement of E2F and RB genes, diverged from other eukaryotes before fungi diverged from metazoans [23,24].

The similarity of yeast SBF/MBF proteins to viral KilA-N proteins has led to the hypothesis that SBF/MBF was introduced into the fungal lineage by virus-mediated horizontal gene transfer [24,33]. This gene transfer event, in which a virus-derived KilA-N/SBF gene became integrated into the genome of the fungal ancestor, may have played a role in the divergence of fungi from metazoans. The functions of viral KilA-N proteins are poorly understood. However, given that SBF/MBF and E2F transcription factors bind similar DNA motifs [24], some viral KilA-N proteins may also bind E2F DNA motifs and activate the expression of E2F target genes in infected cells (Figure 2A). By activating genes important for DNA replication, KilA-N proteins might help create a cellular environment favorable to viral replication in the host [24].

This proposed hijacking of the E2F-RB pathway by a virus-derived SBF-like gene in the fungal ancestor is similar to the mechanism by which the human papilloma virus (HPV) E7 protein hijacks the canonical E2F-RB pathway to cause cervical cancer (Figure 2B). The E7 protein is expressed early after infection and targets RB family members for inactivation [34]. The interaction of E7 with RB releases transcriptionally active E2F to induce the expression of cellular genes important for DNA replication. That is, E7 creates a permissive environment in infected cells allowing HPV to replicate its small, circular genome. On rare occasions, a part of the HPV genome that encodes E7 and another viral oncoprotein, E6, integrates into a host cell genome causing deregulated proliferation and carcinogenesis.

RB homologs in animals and plants actively repress transcription by interacting with and recruiting chromatin modifying enzymes to the promoters of E2F target genes [13,17,27,35,36]. Several of these chromatin modifying enzymes contain a LxCxE amino acid motif that binds to a conserved cleft in the RB pocket domain and allows RB to bind both the chromatin modifiers and E2F factors simultaneously [37]. Cyclin D homologs in animals and plants also have LxCxE motifs that are used to bind and target RB for CDK-mediated phosphorylation. Moreover, the HPV E7 protein, and many other proteins from viruses that infect both animals and plants, also contain LxCxE motifs that are used to disrupt RB-E2F complexes [5,8,38]. Apparently, a number of unrelated DNA viruses have independently evolved LxCxE motif-containing proteins to target RB-E2F complexes as a common strategy for virus replication in both animals and plants [17,39].

## 3. Shared Functions of RB and E2F in Animals and Plants beyond the Mitotic Cell Cycle

### 3.1. Endoreplication

In multicellular animals and plants, E2F and RB homologs also regulate variations of the cell cycle collectively referred to as endoreplication [40]. Endoreplication involves unlinking the S and M phases of the cell cycle so that cells faithfully re-replicate their genomes without cell division. This results in polyploidy and is important for the function of several specialized tissues in animals and plants. Examples of tissues and cell types that undergo endoreplication include plant endosperm and leaves, *Drosophila* nurse and somatic follicle cells, and rodent trophoblast giant cells and liver hepatocytes. In each of these cases, the loss of functional E2F and/or RB homologs significantly impacts endopolyploidy and the normal development of these specialized tissues [41,42,43,44,45,46,47,48]. During endoreplication, RB and E2F homologs have roles beyond the transcriptional regulation of genes important for the G1/S phase transition and the mitotic cell cycle. In some cases, E2F and RB homologs regulate the expression of species-specific genes that control endocycling. In other cases, E2F and RB have transcription-independent functions at sites of endoreplication [40,48,49,50]. Given that endoreplication is restricted to specialized tissues of multicellular eukaryotes, roles for RB and E2F in regulating endoreplication likely evolved independently in animals and plants.

### 3.2. Differentiation

Another shared function of E2F and RB that evolved independently in multicellular animals and plants is the coordination of cell cycle exit with the induction of differentiation programs. This includes roles for RB and E2F in the first step of differentiation: The asymmetric division of stem cells. This asymmetric division gives rise to one daughter cell that maintains “stemness” in a quiescent state and another daughter that proliferates and differentiates into a tissue [13,17,48,51,52,53,54,55]. Genetic experiments in animals and plants have provided numerous examples showing how RB and E2F family members regulate both tissue differentiation and organismal development [56,57]. For example, complete or conditional inactivation of *Rb1* in mice impacts hematopoiesis, neurogenesis, adipogenesis, myogenesis, and osteogenesis [58,59,60,61,62,63,64]. Genetic manipulation of *RB* and *E2F* homologs in *Arabidopsis* has also demonstrated important roles for these proteins in differentiation and the development of plant tissues, including the vascular system, roots, leaves, flowers, and seeds [17,27,36].

The regulation of cell differentiation by E2F and RB homologs in both animals and plants not only involves the transcriptional repression of E2F target genes associated with cell proliferation but also the regulation of fate-determining and tissue-specific genes. For example, E2F and RB directly repress the expression of pluripotent genes, such as *OCT4* and *SOX2* in mammals [51,52,65]. In contrast to their canonical function in transcriptional repression, E2F-RB complexes can also cooperate as transcriptional activators of tissue-specific genes, such as during mouse osteoblast differentiation [60,61]. In *Arabidopsis*, E2F and RB homologs transcriptionally repress tissue-specific genes needed for late stages of seed maturation while E2F simultaneously activates the expression of genes important for cell proliferation [66]. RB homologs in animals and plants also regulate cell differentiation processes by binding to additional, tissue-specific transcription factors. For example, RB regulates erythropoiesis in mice by interacting with the GATA1 transcription factor, which contains an LxCxE motif conserved in other animal GATA1 homologs [67]. A number of tissue-specific transcription factors in *Arabidopsis* also contain LxCxE motifs and cooperate with the sole RB homolog, RBR, to regulate asymmetric division and the development of several distinct plant tissues [17,27,68].

### 3.3. Repeat Silencing

Whole-genome chromatin-association studies have revealed that RB localizes not only to gene promoters, but also to intergenic regions enriched with transposable elements (TEs) in both mammals and plants [69,70]. Mouse models with subtle separation-of-function mutations in RB were used to show that its association with TEs requires neither the pocket domain region that binds E2F family members nor the LxCxE motif-binding region. Instead, RB recruitment to TEs and other repetitive elements requires binding specifically to E2F1 (and not other E2F proteins) through a unique, CDK-resistant mechanism involving the C-terminus of RB [70,71,72]. This non-canonical E2F1-RB complex binds and recruits the EZH2 histone methyltransferase, a component of the polycomb repressive complex 2 (PRC2), to repetitive sequences to promote histone H3 lysine 27 trimethylation (H3K27me3) and transcriptional repression [70]. The absence of this E2F1-specific RB complex substantially reduces H3K27me3 levels, leading to aberrant expression of TEs and other repetitive elements [70]. RB family members also recruit the nucleosome remodeling and deacetylation (NuRD) complex to TEs to help silence TE expression in mice [73]. Significantly, this role in transcriptional silencing may contribute to RB’s tumor suppressive activity; a mutation in RB that impairs silencing at repetitive elements but maintains function in cell cycle regulation predisposes mice to lymphomagenesis [70].

In *Arabidopsis*, approximately one-third of all chromatin-associated RBR is found at TEs, and loss of RBR function results in the aberrant expression of at least some types of TEs [69]. Although a role for E2F factors in recruiting RBR to TEs has not been directly established, the majority of RBR-associated TE sequences contain a consensus E2F DNA-binding motif [69]. This is consistent with the finding that 73% of consensus E2F binding sites in *Arabidopsis* are present in TEs even though TEs make up only 21% of the genome [74]. The mechanism by which RBR silences TEs in Arabidopsis is unknown, but polycomb complexes are well-established, functional partners of RBR in repressing transcription in plants [36]. Moreover, homologs of retinoblastoma-associated proteins 46 and 48 (RbAP46/48), which are components of the NuRD complex, are involved in TE silencing in plants [75]. Taken together, these findings suggest that RB homologs in both mammals and plants participate in TE silencing, perhaps through similar epigenetic mechanisms.

Phylogenomic analyses have revealed that most classes of TEs are widely distributed throughout modern eukarya and therefore likely existed before the emergence of eukaryotes [76,77]. Thus the LECA, like eukaryotic cells today, must have faced the genome destabilizing effects of active TEs. Therefore, it is possible that repressing TE expression was an original core function of E2F and RB homologs in the LECA. In fact, it has been suggested that many E2F DNA binding elements in gene promoters originally arose from TEs inserting into gene regulatory regions [74]. If this is the case, then the TE silencing function of E2F and RB homologs might have evolved first, with the cell cycle regulation function being a later adaptation of the E2F-RB network.

### 3.4. Other Non-Canonical Functions

Mammalian RB and E2F homologs are also implicated in other cellular processes, including functions that are independent of their abilities to regulate transcription. For example, RB can regulate apoptosis by interacting with BAX at mitochondria, and can regulate the initiation of DNA synthesis by interacting with ORC and MCM proteins at origins of replication [78,79,80,81,82]. RB homologs in both *Drosophila* and mammals also directly regulate chromatin condensation during mitosis by interacting with a condensin complex [83,84,85,86]. In mammals, this function depends on the E2F1-specific RB complex, is independent of LxCxE-motif binding, and is distinct from the EZH2-dependent mechanism used by E2F1 and RB to silence TEs [21,86]. In mammals, RB family members also appear to be general regulators of chromatin structure by regulating histone H4K20 methylation in an E2F-independent manner [85,87,88,89]. At present, however, there is little evidence that homologs of RB or E2F in plants have transcription-independent functions in regulating programmed cell death, DNA replication, or chromatin condensation.

## 4. Homologs of E2F and RB in Mammals and Plants Directly Regulate DNA Double-Strand Break Repair

Another shared function of RB and E2F in mammals and plants is a direct role in promoting DNA DSB repair at sites of damage. E2F1 and RB (in mammals) and E2FA and RBR (in plants) each form foci that overlap with markers of DNA DSBs, such as γH2AX and BRCA1 [90,91,92,93,94,95]. Localization of mammalian E2F1 and RB to DSBs requires phosphorylation of E2F1 by the ATM kinase and phospho-specific binding to a BRCT domain in the TopBP1 protein [91,94,95]. RB participates in this process by stabilizing the interaction between phosphorylated E2F1 and TopBP1 at sites of damage (Figure 3). ATM kinase activity is also required for plant E2FA and RBR to localize to DSBs, but whether E2FA phosphorylation or binding to TopBP1 is involved has not been tested [92,93].

In both mammals and plants, the localization of E2F and RB homologs to DSBs is associated with the promotion of repair by HR and the maintenance of genome integrity [90,91,92,95,96,97,98,99,100]. Some early events in the DNA damage response, such as γH2AX foci formation, are unaffected by the absence of RB or E2F homologs. On the other hand, recruitment of the RAD51 DNA recombinase to sites of damage is significantly impaired in the absence of either RB/RBR or E2F1/E2FA [90,91,92,95]. In mammals, loading of the MRE11-RAD50-NBS1 (MRN) complex at DNA breaks is also compromised when RB or E2F1 are depleted or when E2F1 is mutated [91,99]. Others have shown that RB is also important for the recruitment of CtIP, a protein that works with MRN in the DNA end resection step of DSB repair [98]. Whether RBR and E2FA promote the recruitment of *Arabidopsis* MRN or CtIP to sites of damage has not been examined, but it would be consistent with impaired RAD51 foci formation in *RBR* mutant cells [90,92]. This direct role for mammalian and plant homologs of E2F and RB in HR repair extends to meiotic recombination [95,101].

In mammals, the regulation of DNA repair by E2F1 and RB involves remodeling chromatin structure at DNA breaks to enhance access to the repair machinery and to facilitate DNA end processing [95,99]. The mechanism by which E2F1 and RB remodel chromatin at DNA breaks shares similarities with how they regulate gene transcription. In fact, E2F1 and RB recruit some of the same chromatin modifying enzymes, e.g., the histone acetyltransferases p300 and CBP, and the chromatin remodeler BRG1, to both DSBs and gene promoters [60,61,95,99,102,103,104,105,106,107]. Recruitment of p300 and CBP to DSBs involves an interaction between the bromodomains of p300 and CBP and a motif on E2F1 that is acetylated in response to DNA damage (Figure 3) [99]. Recruitment of p300/CBP to DSBs by E2F1 and RB leads to the acetylation of histone H3 at lysines 18 (H3K18ac) and 56 (H3K56ac) in chromatin flanking DSBs. These findings establish a link between E2F1 acetylation and histone acetylation at DSBs and reveal a novel E2F1 reader function for the bromodomains of p300 and CBP.

BRG1 recruitment to DSBs also involves an association with the TopBP1-E2F1-RB complex, likely through the well-established interaction between BRG1 and RB [102,106]. The histone H3 acetylation marks deposited by p300/CBP may also act as docking sites for the bromodomain of BRG1 to facilitate its recruitment and/or function at DSBs. Recent reviews have highlighted how histone acetyltransferases and nucleosome remodeling complexes cooperate to modify chromatin structure at sites of DNA damage to facilitate repair [108,109,110]. In particular, p300/CBP-mediated histone acetylation at sites of damage cooperates with BRG1 to reduce nucleosome density and allow efficient DNA end resection [95,99]. E2F1 and RB also indirectly promote the recruitment of the Tip60 acetyltransferase to induce histone H4 acetylation, another histone mark associated with efficient DSB repair [99,111]. Although it has been proposed that *Arabidopsis* E2FA and RBR also promote HR repair by remodeling chromatin structure at sites of damage, direct regulation of chromatin structure flanking DSBs by plant E2F and RB homologs has not yet been reported [112].

Recruitment of mammalian E2F1 to DSBs is independent of its DNA binding and transcriptional activation domains, as an N-terminal 83 amino acid fragment of E2F1 is sufficient to localize a fused GFP reporter to DSB foci [94]. In contrast, mutating the E2F1 ATM/ATR-mediated phosphorylation site (serine 31 in humans and serine 29 in mice) abolishes binding to TopBP1 in response to DNA damage and blocks the recruitment of E2F1 to DSBs [94,95]. A mouse knock-in model was developed in which this site of phosphorylation was mutated to alanine (S29A) [113]. Like *E2f1^-/-^* knockout mice, *E2f1^S29A/S29A^* knock-in mice are viable and appear normal. The expression of E2F target genes, including those involved in cell cycle regulation, DNA damage response, and DNA repair, are largely unaltered in cells and tissues from *E2f1^S29A/S29A^* mice [95,113]. However, the E2F1 S29A mutation prevents recruitment of E2F1, RB, BRG1, and p300/CBP to DSB sites and also impairs the recruitment of repair proteins like NBS1, Mre11, and Rad51 [95,99]. Thus, the *E2f1^S29A/S29A^* knock-in mouse model provides an experimental tool that specifically impairs the functions of E2F1, RB, BRG1, and p300/CBP at sites of DNA damage but not their functions in regulating cell cycle progression and other processes. It is notable that *E2f1^S29A/S29A^* knock-in mice are not significantly prone to developing spontaneous tumors but they are more prone to ultraviolet radiation-induced skin carcinogenesis [113]. *E2f1^S29A/S29A^* mice are also hypersensitive to ionizing radiation (IR) and have reduced fertility, phenotypes often associated with defective HR. Interestingly, loss of E2FA and RBR function in *Arabidopsis* also causes hypersensitivity to DNA damage and infertility [90,92,93,101].

Another *E2f1* knock-in allele, termed 3KR, was developed to prevent acetylation of E2F1 and the interaction between E2F1 and the bromodomains of p300 and CBP [99]. In cells from *E2f1^3KR/3KR^* mice, E2F1 and RB are still recruited to DSB sites, but p300, CBP and BRG1 are not. Further, *E2f1^3KR/3KR^* cells display DNA repair defects like those observed in other *E2f1* mutant cells [99]. Finally, like *E2f1^−/−^* knockout and *E2f1^S29A/S29A^* knock-in mice, *E2f1^3KR/3KR^* knock-in mice are also hypersensitive to IR, indicating that the recruitment of chromatin modifying enzymes to sites of DSBs is a critical function of E2F1 and RB [99].

Given the similarities between mammalian and plant E2F and RB homologs in their abilities to localize to sites of DSBs and to directly promote efficient HR repair, it is possible that this transcription-independent function of E2F and RB was present in the LECA, before the divergence of plants and animals. Nonetheless, RB and E2F homologs have been demonstrated to localize to DNA damage foci only in humans, mice, *Arabidopsis*, and tobacco [90,91,92,93,94,95]. The N-terminal motif in E2F1 that is phosphorylated by ATM and is critical for E2F1 and RB recruitment to DSBs (SSQ), is conserved in other placental mammals but not in marsupials, monotremes, other vertebrates, or model invertebrates (Figure 4). This suggests that either E2F and RB homologs are not recruited to DSBs in animals outside the placental mammals or that E2F and RB homologs are recruited to DSBs in other eukaryotes through a different mechanism.

Interestingly, E2FA in *Arabidopsis*, like E2F1 in placental animals, contains an “SSQ” motif in its N-terminus. However, whether this motif is phosphorylated by ATM or is involved in the recruitment of E2FA and RBR to sites of DNA damage in *Arabidopsis* is unknown. Regardless, the lack of conservation of this phosphorylation motif in other eukaryotes suggests that the ability of E2F and RB homologs to localize to sites of DNA damage in placental mammals and plants evolved independently. If so, then this independently acquired trait allowing E2F and RB to directly promote DSB repair is an example of convergent evolution at the molecular level.

## 5. Conclusions

RB and E2F are fundamental components of the cell cycle machinery in most eukaryotes. In both animals and plants, RB and E2F homologs have evolved additional, non-canonical functions beyond cell cycle regulation. Dysregulation of the RB-E2F axis is a hallmark of human cancer and occurs through multiple mechanisms. In some cancers, the *RB1* gene is deleted or mutated such that both the canonical and non-canonical functions of RB are lost. These functions include a direct role for RB, in partnership with E2F1, in repairing DNA DSBs in the context of chromatin. Further research is needed to understand how loss of this repair function and other non-canonical functions contribute to the progression of cancers that have lost RB. Yet, despite these gaps in understanding, the finding that RB is important for HR repair reveals a potential vulnerability: Tumors lacking RB might be exploited therapeutically by treatments similar to those used for other cancers with impaired HR repair (e.g., PARP inhibitors) [95,98,114]. For cancers lacking *RB1* mutations, although the canonical functions of RB might still be disrupted through abnormal CDK activity or association with viral oncoproteins, intact non-canonical functions of RB in DNA repair might foster resistance to conventional cancer therapies. For example, RB is recruited to DSB sites and contributes to efficient DNA repair in the human osteosarcoma cell line U2OS, which expresses a normal RB protein but lacks the p16 CDK inhibitor [95,99]. Consequently, exploiting this repair mechanism by inhibiting the function of RB in HR repair might sensitize RB-positive tumors to DNA DSB-inducing therapies. Indeed, blocking E2F1 and RB recruitment to DNA breaks may contribute to the efficacy of ATM inhibitors currently being investigated in the clinic [95,115]. Another possible approach for disrupting RB function in DSB repair would be to prevent the E2F1- and RB-dependent recruitment of p300 and CBP to sites of damage, using inhibitors of the bromodomains of p300 and CBP [99]. Continuing to refine our understanding of the non-canonical functions of RB and E2F family members may reveal additional approaches for the treatment of both RB positive and negative cancers. Given that many of these non-canonical functions of RB and E2F are present across Eukarya, studies in appropriate model organisms should play an important role in these investigative efforts.

## Figures and Tables

**Figure 1 cancers-13-00934-f001:**
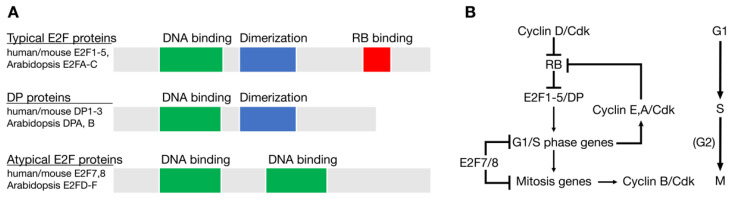
Conservation of E2F and RB in the Last Common Eukaryotic Ancestor (LECA). (**A**) Structure of the three types of E2F family member proteins. (**B**) The cell cycle regulatory network of the LECA based on genomic sequence comparisons across eukaryotic groups.

**Figure 2 cancers-13-00934-f002:**
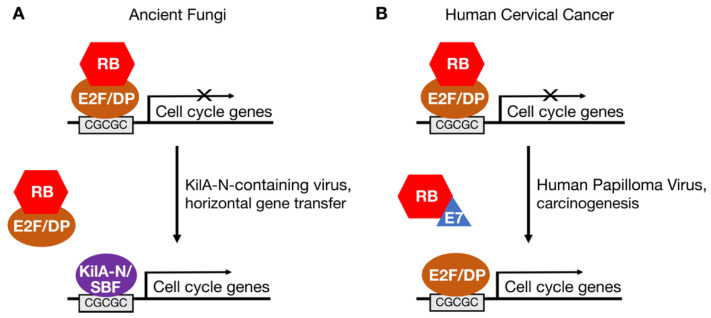
Hijacking of the RB-E2F pathway by viral proteins in the fungal ancestor and in cervical cancers. (**A**) SBF transcription factors in fungi may have arisen through a horizontal gene transfer event involving a virus encoding a KilA-N-related protein. (**B**) Human papilloma virus (HPV) encodes the E7 oncoprotein, which inhibits RB and activates endogenous E2F transcription factors to drive expression of cellular genes needed for viral replication. E7 is one of the viral genes that is integrated into the genome of most cervical cancer cells.

**Figure 3 cancers-13-00934-f003:**
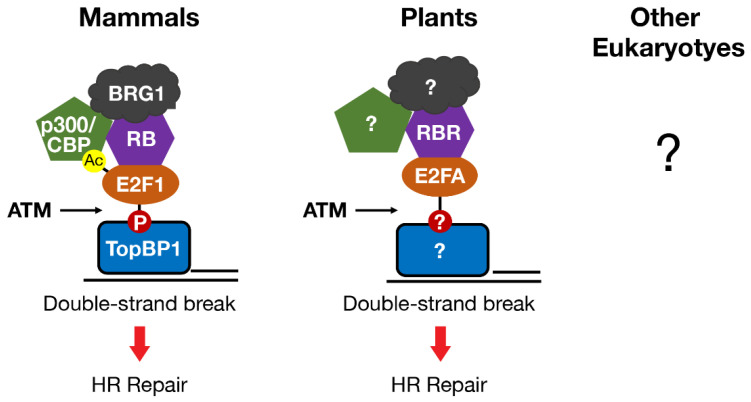
E2F and RB homologs in mammals and plants localize to DNA double-strand breaks in an ATM-dependent manner to directly promote repair by homologous recombination. In mammals, E2F1 post-translational modifications regulate protein–protein interactions that localize E2F1-RB to double-strand breaks and recruit the chromatin modifying enzymes p300/CBP and BRG1. Whether E2F and RB homologs directly regulate DNA repair by similar mechanisms in other eukaryotes is unknown.

**Figure 4 cancers-13-00934-f004:**
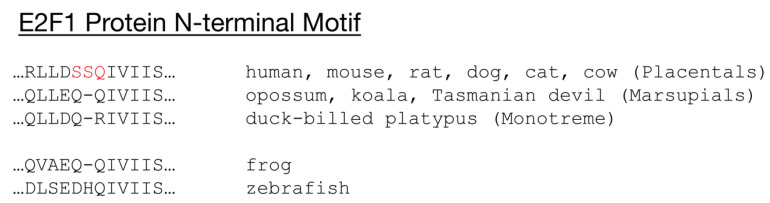
The E2F1 protein N-terminal motif phosphorylated by ATM and required for E2F1 and RB location to DNA damage is not conserved beyond placental mammals.

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
