# Peer review of "Direct Regulation of DNA Repair by E2F and RB in Mammals and Plants: Core Function or Convergent Evolution?"

_cancers, 2021, doi:10.3390/cancers13050934_

Round 1
Reviewer 1 Report
In the manuscript titled with "Direct Regulation of DNA Repair by E2F and RB in Mammals and Plants: Core Function or Convergent Evolution?", authors gave a good summary about the function of E2F and RB in DNA damage repair, and discussed about if the function of E2F and RB homologs in DNA damage repair is evolutionally conserved in mammals and plants.
The manuscript is well written, and I have following specific minor comment:
1. In line 63, while talking about 3 kinds of E2F genes, it will be better if a schematic diagram is included to show the similarity and difference between them, domains and protein interaction region can also be included in the diagram, which will help the readers to understand the following mechanism part too.
overall, this is a good review about the function of RB and E2F in DNA damage repair.
Author Response
Reviewer 1
Suggestions for Authors
In the manuscript titled with "Direct Regulation of DNA Repair by E2F and RB in Mammals and Plants: Core Function or Convergent Evolution?", authors gave a good summary about the function of E2F and RB in DNA damage repair, and discussed about if the function of E2F and RB homologs in DNA damage repair is evolutionally conserved in mammals and plants.
The manuscript is well written, and I have following specific minor comment:
1. In line 63, while talking about 3 kinds of E2F genes, it will be better if a schematic diagram is included to show the similarity and difference between them, domains and protein interaction region can also be included in the diagram, which will help the readers to understand the following mechanism part too.
A new schematic diagram showing the structural domains of the three different types of E2F proteins is now included as Figure 1A.
overall, this is a good review about the function of RB and E2F in DNA damage repair.
Reviewer 2 Report
The review proposed by Manickavinayaham et al. aims at putting in perspectives the commonalities of E2F and RB non-canonical functions in plant and in animal. The review is in overall interesting and easy to reads. Below are suggestions which will I hope help the authors to further improve their review.
1) The title announce a focus on DNA repair but this point is presented in the last paragraph of the manuscript. The length of this last part being relatively small, the title is for me not adequately chosen as DNA repair is a process described among other. The authors should either extend the review on DNA damages per se, or change their title.
2) the part on DNA damages does not describe the mechanistic implication of gamma H2AX. I would like the authors to add a paragraph on this point including the interactions of this epiG modification in the context of E2F and RB signalling (if any).
3) a last part to make a summary and perspectives would be nice for this review. At the moment such part is missing
4) The panel 2A on ancient fungi should be to my opinion redrawn. I perhaps missed something but I think it is misleading to propose RB/E2F that repress cell cycle progression and SBF that promote cell cycle progression with RB/E2F apart from the promoter. The proposition was more that horizontal gene transfer led to the replacement of the RB complex by the SBF complex for the regulation of the cell cycle gene. In the text line 136-146 the authors should state if viral Hijacking of the RB/E2F do occurs, and then propose alternative such competition with endogenous RB/E2F complex. The authors should also discuss in this part their conclusion in term of evolution history of the RB/EF complex in plants.
5) The DNA sequences recognized by E2F, KilA and and SBF should be better described. The authors state several times that the different TF could recognize similar DNA motif . I think that describing directly the different transcription factor binding sites and their potential sequence homologies will be more rigorous. A dedicated figure could be perhaps proposed. In addition, line 116, the authors question if Kil A could activate the same target genes that E2F because of similar binding sites. A reference should be provided to support this hypothesis.
6) Similarly, line 120-126: a description of the conservation of structural features or properties between E7 and RB proteins should be proposed.
Author Response
Reviewer 2
The review proposed by Manickavinayaham et al. aims at putting in perspectives the commonalities of E2F and RB non-canonical functions in plant and in animal. The review is in overall interesting and easy to reads. Below are suggestions which will I hope help the authors to further improve their review.
1) The title announces a focus on DNA repair but this point is presented in the last paragraph of the manuscript. The length of this last part being relatively small, the title is for me not adequately chosen as DNA repair is a process described among other. The authors should either extend the review on DNA damages per se, or change their title.
The review is structured into a short introduction (31 lines), a section on the evolutionary history of RB and E2F (95 lines), a section on non-canonical functions of RB and E2F (101 lines), and a final section focusing on the role of RB and E2F in DNA repair (now 119 lines). We feel that the sections on evolution and non- canonical functions are critical for setting up the final, largest section describing RB and E2F function in DNA repair and how this ability evolved in both mammals and plants. We therefore request that the title remain the same to highlight the main, more novel findings presented in the review.
2) the part on DNA damages does not describe the mechanistic implication of gamma H2AX. I would like the authors to add a paragraph on this point including the interactions of this epiG modification in the context of E2F and RB signaling (if any).
A new paragraph better describing the relationship between RB-E2F and H2AX and other DNA damage response factors has been added to the revised manuscript (lines 271-284).
3) a last part to make a summary and perspectives would be nice for this review. At the moment such part is missing
A concluding paragraph with summary and perspectives has been added to the revised manuscript.
4) The panel 2A on ancient fungi should be to my opinion redrawn. I perhaps missed something but I think it is misleading to propose RB/E2F that repress cell cycle progression and SBF that promote cell cycle progression with RB/E2F apart from the promoter. The proposition was more that horizontal gene transfer led to the replacement of the RB complex by the SBF complex for the regulation of the cell cycle gene. In the text line 136-146 the authors should state if viral Hijacking of the RB/E2F do occurs, and then propose alternative such competition with endogenous RB/E2F complex. The authors should also discuss in this part their conclusion in term of evolution history of the RB/EF complex in plants.
Figure 2 has been redrawn as suggested. It has been simplified to focus on the horizontal gene transfer event that introduced the SBF gene into ancient fungi and the analogy with the carcinogenesis event that occurs during the development of human cervical caused by HPV.
5) The DNA sequences recognized by E2F, KilA and and SBF should be better described. The authors state several times that the different TF could recognize similar DNA motif . I think that describing directly the different transcription factor binding sites and their potential sequence homologies will be more rigorous. A dedicated figure could be perhaps proposed. In addition, line 116, the authors question if Kil A could activate the same target genes that E2F because of similar binding sites. A reference should be provided to support this hypothesis.
The DNA sequence that is recognized by E2F and SBF (CGCGC) has now been incorporated into figure 2. A reference has now been added (line 122) to support the hypothesis as suggested.
6) Similarly, line 120-126: a description of the conservation of structural features or properties between E7 and RB proteins should be proposed.
RB and E7 do not share structural features. E7 shares structural features (LxCxE motif) with other RB binding proteins. This is described in the next paragraph, in particular lines 153-155.
Reviewer 3 Report
The review manuscript submitted by Manickavinayaham and co-authors entitled “Direct Regulation of DNA Repair by E2F and RB in Mammals and Plants: Core Function or Convergent Evolution?” is focused on the non-canonical functions of RB and E2F proteins in maintaining eukaryotic genome integrity. Based on their own results and the literature data the authors discuss the parallels between mammalian and plant E2F and RB homologs with respect to their ability to localize to DSB and promote homologous recombination repair. The authors also try to answer the question of whether this non-canonical function of E2F and RB has been present before the divergence of plants and animals or it has evolved independently in mammals and plants. They present evidences supporting their conclusion that the shared function of mammalian and plant E2F and RB homologs allowing them to promote efficient DSB repair is an example of convergent evolution at the molecular level.
The manuscript is well written and very comprehensive. The figures are easy to perceive and correctly exemplify the key discussion points.
The manuscript is very detailed and affects different aspects of the molecular functions that E2F and RB play in the cell. Therefore, for better understanding by the readers the authors are advised to finalize the review with a short subheading highlighting their key conclusions.
Author Response
Reviewer 3
The review manuscript submitted by Manickavinayaham and co-authors entitled “Direct Regulation of DNA Repair by E2F and RB in Mammals and Plants: Core Function or Convergent Evolution?” is focused on the non-canonical functions of RB and E2F proteins in maintaining eukaryotic genome integrity. Based on their own results and the literature data the authors discuss the parallels between mammalian and plant E2F and RB homologs with respect to their ability to localize to DSB and promote homologous recombination repair. The authors also try to answer the question of whether this non-canonical function of E2F and RB has been present before the divergence of plants and animals or it has evolved independently in mammals and plants. They present evidences supporting their conclusion that the shared function of mammalian and plant E2F and RB homologs allowing them to promote efficient DSB repair is an example of convergent evolution at the molecular level.
The manuscript is well written and very comprehensive. The figures are easy to perceive and correctly exemplify the key discussion points.
The manuscript is very detailed and affects different aspects of the molecular functions that E2F and RB play in the cell. Therefore, for better understanding by the readers the authors are advised to finalize the review with a short subheading highlighting their key conclusions.
A concluding paragraph has now been added to the revised manuscript (lines 381-410).
Round 2
Reviewer 2 Report
All comments were taken into account by the authors